# Synthesis and Study of Physical and Mechanical Properties of Urethane-Containing Elastomers Based on Epoxyurethane Oligomers with Controlled Crystallinity

**DOI:** 10.3390/polym14112136

**Published:** 2022-05-24

**Authors:** Alexey Slobodinyuk, Vladimir Strelnikov, Nadezhda Elchisheva, Dmitriy Kiselkov, Daria Slobodinyuk

**Affiliations:** Institute of Technical Chemistry Ural Branch of the Russian Academy of Sciences, Ac. Korolev 3, 614130 Perm, Russia; svn@itcras.ru (V.S.); env-1981@mail.ru (N.E.); dkiselkov@yandex.ru (D.K.); selivanovadg@gmail.com (D.S.)

**Keywords:** curing agent, epoxyurethane oligomer, oligotetramethylene oxide diol, synthesis, bromination, Gabriel reaction

## Abstract

The influence of the molecular weight of oligoamine, oligoether, and the type of diisocyanate on the physical and mechanical properties of elastomers with urethane hydroxyl hard segments was studied. For this purpose, oligoetherdiamines with molecular weights ~1008 and ~1400 g mol^−1^ were synthesized by a three-stage method. Epoxyurethane oligomers were synthesized according to a two-step route with an oligodiisocyanate as an intermediate product. A series of 12 elastomers with controlled crystallinity were synthesized from these elastomers and amines. The deformation and strength properties of the elastomers were studied.

## 1. Introduction

Smart materials are the next generation of functional materials, after natural, man-made, and synthetic materials [1,2]. These high-tech materials combine the advantages of structural and functional materials [3,4,5].

Shape memory polymers (SMPs) are a type of smart materials that can recover from a deformed temporary shape to a “memorized” permanent shape in response to an external environmental stimulus, for example, light, temperature, solvent, pH, electrical or magnetic field [6,7,8,9,10]. Shape-memory polymers can be both thermoplastic and thermosetting ones. The mechanical properties, thermal stability, fixation ratio, and recovery rate of the thermosets are higher than those of the thermoplastics [11]. At present, these polymers are used in aerospace structures and biomedical devices [12,13,14,15,16].

Recently, segmented polyurethanes have been increasingly considered as shape memory materials. This class of polymers is characterized by a high shape recovery ability, controlled shape recovery temperature, high stress–strain properties, wear resistance, and biocompatibility.

In general, segmented urethane-containing elastomers (SUE), such as polyurethanes and polyurethane ureas, can be considered to be a unique class of polymers. The structure and properties of these polymers can be regulated over a wide range [17]. Segmented polyurethanes (SPU) and polyurethane ureas (SPUM) are block copolymers with macromolecular chains consisting of the alternating soft and hard segments, SS and HS, respectively. Hard segments are formed as a result of the reaction of a diisocyanate with low molecular weight chain extenders, such as diamines and diols. The soft segment structure is determined by the oligomers used in the synthesis of an SUE [18].

The difference in the polarity of soft and hard segments results in the microphase separation followed by the formation of hard domains [19,20,21]. In the microdomains, the hydrogen bonds play an important role in the stabilization of the hard phase structure [22]. Solid domains act as a reinforcing filler. These domains are in the nodes of a specific physical network stable in a wide range of positive temperatures [23].

It should be noted that the shape memory effect of polyurethane elastomers is largely dependent on the degree of crystallinity and hydrogen bonds of the polymer, as well as its microphase separation [24]. These parameters can be varied by using polyesters with different molecular weights, different types of diisocyanates, etc.

Despite all the advantages of polyurethanes and polyurethane ureas, the deformation and strength properties of these elastomers depend on humidity. This is due to the ability of the terminal isocyanate groups, present in oligodiisocyanates, to react with water. To solve this problem, isocyanate groups should be “blocked” by some reactive substance. From this point of view, the most promising reactants are epoxy alcohols, for example, 2,3-epoxy-1-propanol [25]. In this case, the isocyanate group of the oligodiisocyanate reacts with the hydroxyl group of 2,3-epoxy-1-propanol to provide an epoxyurethane oligomer. In addition, the toxicity of isocyanate terminal compounds is substantially reduced. For the elastomers based on these oligomers, no deterioration of the stress–strain properties is observed when subjected to humidity. Amines, anhydrides of dicarboxylic acids, carboxyl-containing compounds, and imidazoles are used as curing agents for epoxyurethane oligomers [26,27,28,29,30,31].

The epoxyurethane elastomers are characterized by high dielectric properties and mechanical characteristics, and, not least, their properties are independent of humidity.

The elastomers, synthesized using polyester-based epoxyurethane elastomers, are crystallizable [25] and can be used as shape memory materials. However, the use of these materials at low temperatures is limited by a high glass transition temperature of −30–40 °C.

The glass transition temperature of polyether-based epoxyurethane elastomers is up to −72–74 °C. Thus, at a low temperature, it is advisable to use this class of elastomers. However, these elastomers are non-crystallizable, and this property limits the use of these polymers as shape memory materials.

In our opinion, for the preparation of crystallizable polyether-based elastomers, linear oligomer curing agents should be used.

In literature, a wide variety of synthetic methods for oligoethylene glycol diamine have been described, and the synthesis of oligopropylene glycol diamine (trademark Jeffamine) is also well-developed. Such oligomers can be prepared from oligoglycols by a variety of synthetic routes. The most common synthetic route is a three-step procedure that includes mesylation, azidation, and reduction [32]. In addition, in [33], another preparation method for oligoether diamines is described. According to this procedure, amino terminated oligomers can be prepared via the two-step route: substitution of oligoglycol hydroxyls with phthalimide moieties, followed by the hydrazinolysis reaction. Synthetic methods for oligoetherdiamines, including the step of catalytic amination under elevated pressure and temperature conditions, have been also studied [34,35]. As reported by J. M. Harris and colleagues, oligoethylene glycol diamine was synthesized via oxidation of oligoethylene glycol followed by reduction of carbonyl compound [36]. In our opinion, the elastomers cured with these amines are likely to be non-crystallizable. In the first case, when using methyl branched Jeffamine, it is due to the steric hindrance, and in the second—to high oxygen content.

In the present paper, the elastomers cured with oligoamine, synthesized from oligotetramethylene oxide diol, are studied. The authors consider this oligoamine to be the most promising curing agent for the preparation of crystallizable elastomers based on epoxyurethane oligomers.

The aim of the present study was the preparation and characterization of cold-resistant crystallizable elastomers. Four epoxy-terminated oligomers based on oligotetramethylene oxide diols of different molecular weights were synthesized. A novel synthetic method for oligoamines was developed. Two amino terminated oligomers with molecular weights of 1000 and 1400 were used as curing agents. The elastomers based on oligoethers and oligoamines were synthesized and characterized.

## 2. Materials and Methods

### 2.1. Materials and Synthesis

#### 2.1.1. Materials

Phosphorus tribromide 99% (Sigma-Aldrich Co., St. Louis, MO, USA), Phthalimide potassium salt 98% (Sigma-Aldrich Co., St. Louis, MO, USA), Hydrazine monohydrate 98% (Sigma-Aldrich Co., St. Louis, MO, USA), 2,4-toluene diisocyanate (TDI) (BASF, Ludwigshafen, Germany), isophorone diisocyanate (IPDI) (Evonik Chemistry Ltd., Essen, Germany), oligotetramethylene oxide diol (OTMO; BASF, Ludwigshafen, Germany) with M_n_ ~ 1008 g∙mol^−1^, M_n_ ~ 1400 g∙mol^−1^, glycidol (grade pure, 99.0%, Research Institute of Polymer Materials, Perm, Russia), Dibutyltin dilaurate (grade pure, 99.8%) were used without purification.

#### 2.1.2. Synthesis of OTMO-Diamines

The three-step synthetic route for preparation of OTMO-diamines is presented in Figure 1.

Nucleophilic substitution of polyfurite hydroxyls with bromine was carried out using phosphorus tribromide. This approach excludes the use of toxic thionyl chloride, as in the methods of polyethylene glycol halogenation described in [37]. Bromination of polyfurite was carried out in a round-bottomed three-necked flask equipped with a mechanical stirrer and a thermometer. First, a weight of polyfurite (23 mmol, M_n_ ~ 1008 g∙mol^−1^, M_n_ ~ 1400 g∙mol^−1^) was dissolved in chloroform (50 mL), and the solution was introduced into the flask and cooled to 0 °C. Then phosphorus tribromide (1.82 mL, 19 mmol) was added to the solution dropwise, and the reaction mixture was heated to room temperature and then boiled for 5 h. After the reaction was completed, reaction mixture was cooled and added to a saturated aqueous sodium hydrocarbonate solution (50 mL). The organic phase was washed with water (3 × 50 mL) and dried over anhydrous magnesium sulfate. The excessive solvent was removed at a rotary evaporator. As a result, OTMO-dibromides were obtained in high yields (95–98%).

In the second step, for preparation of OTMO-diamines, potassium phthalimide (24.5 g, 13.2 mmol) was added to a solution of an OTMO-dibromide (22 mmol, M_n_ ~ 1134 g∙mol^−1^, M_n_ ~ 1526 g∙mol^−1^) in DMF (180 mL) and the reaction mixture was stirred under argon at 120 °C for 5 h. Then the solution was cooled and filtered, and DMF was evaporated in vacuum. For purification, the product was dissolved in DCM, filtered, and the excess solvent was removed at a rotary evaporator. OTMO-diphtalimides were obtained in the form of a white viscous mass in high yields (80–85%).

At the final step, hydrazine hydrate (180 mmol, 8.75 mL) was added to OTMO-diphtalimide (18 mmol) dissolved in absolute ethanol (100 mL). The reaction mixture was stirred at reflux under argon atmosphere for 5 h. Then the solution was cooled and filtered, and ethanol was evaporated under vacuum. For purification, the product was dissolved in DCM and filtered, and the excess solvent was removed at a rotary evaporator. The yield of OTMO-diamines was 50–58%.

#### 2.1.3. Synthesis of Epoxyurethane Oligomers

A two-step synthetic route for epoxyurethane oligomers via oligodiisocyanate formation is shown in Figure 2.

The pre-synthesized polyesters were dried at 90 °C for 7 h. Oligodiisocyanates were obtained via interaction of oligodiols and diisocyanate (NCO/OH = 2.03) at constant temperature of the reaction mixture of 1 h at 60 °C and 80 °C at stirring for 6 h. The content of NCO groups in the prepolymers was determined by titration with n-butylamine (standard method ASTM D 2572-97). Then the reaction mixture was cooled to 40 °C, and the catalyst, di-n-butyl tin dilaurate, and the calculated amount of glycidol were added. The catalyst amount was 0.03 wt.% of the reaction mixture. Then the reaction mixture was heated to 70 °C and stirred for 8 h. The completeness of the reaction was confirmed by IR spectroscopy. No band at 2270 cm^−1^, characteristic of isocyanate group [38], was observed in the IR spectra of the reaction products. The content of free epoxy groups was determined according to the technique described in [39]. The composition and properties of the synthesized oligomers are summarized in Table 1.

#### 2.1.4. Polymer Synthesis

At the final synthetic step, epoxyurethane oligomers were mixed with a synthesized curing agent for 10 min under vacuum (1–2 kPa) at 40 °C. The resulting reaction mixture was cured for 48 h at 30 °C. Cure monitoring by FTIR was used to determine the required cure time. Disappearance of the absorption band at 910 cm^−1^ indicated the completeness of the epoxy group conversion [40]. The synthetic route is demonstrated in Figure 3.

The sample compositions are provided in Table 2.

### 2.2. Methods

#### 2.2.1. ^1^H- and ^13^C-NMR Spectroscopy

^1^H and ^13^C NMR spectra were recorded using a Bruker, Moccow, Russia) Avance Neo III spectrometer (^1^H: 400 MHz, ^13^C: 75 MHz); tetramethyl silane was used as an internal standard. NMR chemical shifts were calibrated using the deuterium signal of CDCl_3_ (7.26 ppm for ^1^H and 77.16 ppm for ^13^C).

#### 2.2.2. Elemental Analysis

Elemental analysis was carried out using a LECO, Moscow, Russia CHNS-932 analyzer.

#### 2.2.3. Gel Permeation Chromatography

The molecular mass of the oligomers obtained was determined by gel permeation chromatography using an ULTIMATE 3000 chromatograph (Dionix Thermo Scientific, Moscow, Russia) equipped with a RefractoMax 521 refractometric detector according to [41].

#### 2.2.4. FTIR Spectroscopy

FTIR spectra in the area of carbonyl valence vibrations (between wave numbers ν = 1600 and 1760 cm^−1^) of the investigated samples were recorded using an IFS-66/S spectrometer (Bruker, Moscow, Russia) with spectral resolution of 1 cm^−1^. The spectra were normalized with respect to the band at 2860 cm^−1^, corresponding to symmetric vibrations of aliphatic –CH_2_ groups [42].

#### 2.2.5. Differential Scanning Calorimetry (DSC)

Endothermic and exothermic effects in the samples within the temperature range from −100 °C to +100 °C were recorded using a Mettler Toledo MDSC Q100 calorimeter (Mettler Toledo, Moscow, Russia). Heating and cooling rates were 5 K min^−1^.

#### 2.2.6. Mechanical Tests

Mechanical tests of specimens of the materials obtained were performed with an Instron 3365 (Moscow, Russia) testing machine at the extension velocity υ = 0.417 s^−1^ and a temperature of 25 ± 1 °C by the standard procedure. The following characteristics were measured: the nominal strength σk (MPa), i.e., the maximal stress per initial specimen cross section; the relative critical strain ε_k_ (%); the nominal elastic modulus E_100_ (stress at the relative strain ε = 100%). The synthesized polymer was subjected to 5 tests.

## 3. Results

This section is divided into subheadings. It should provide a concise and precise description of the experimental results, their interpretation, as well as the experimental conclusions that can be drawn.

### 3.1. NMR Spectra of Functionalized Oligotetramethylene Oxides

The transformations of the terminal fragments of the initial polyfurites were confirmed by NMR data (Figure 4a–d). In the ^1^H NMR spectra, the protons of the methylene groups of the initial polyfurites (M_n_ ~ 1008 g∙mol^−1^, M_n_ ~ 1400 g∙mol^−1^) are observed at 1.51–1.62 (O-CH_2_-C*H*_2_-C*H*_2_-CH_2_-O (a)) and 3.31–3.39 (O-C*H*_2_-CH_2_-CH_2_-C*H*_2_-O (b)) ppm, respectively. The triplet at 3.56 (c) ppm can be attributed to the terminal hydroxy methylene groups. The signals of the two hydroxy group protons appear at 2.31 ppm (M_n_ ~ 1008 g∙mol^−1^) or at 2.60 ppm (M_n_ ~ 1400 g∙mol^−1^). Upon the substitution of hydroxy groups with bromine, these signals disappeared. In addition, the signals attributed to the protons of the methylene groups in the vicinity of halogen atoms were found to be shifted. Further, the substitution of bromine with phthalimide groups resulted in the appearance of phthalimide proton signals in a weak field region (δ = 7.63 ppm and 7.76 ppm), and again a shift of the methylene proton signals in the vicinity of the substituent, in this case, phthalimide, is observed. In the next step, in OTMO diamine spectra, there are the peaks attributed to the protons of two amino groups (δ = 2.87 ppm for M_n_ ~ 1008 g∙mol^−1^ and δ = 2.67 ppm for M_n_ ~ 1400 g∙mol^−1^). The strong field shift of the signals of the functionalized methylene groups is observed.

In ^13^C NMR spectra, at each step of the synthetic route, a shift of the signals of the terminal methylene carbon (c) and (d) can be distinguished (Figure 4b,d). In addition, weak field signals (δ = 123, 132, 134, and 168 ppm) appear in the spectra of OTMO-diphtalimides. These signals disappear after the aminolysis is completed.

Number-average molecular weight (M_n_) was determined from ^1^H NMR spectroscopy by comparing the integration of the end-group proton resonances to the repeating unit proton resonances. The results are presented below.

OTMO-dibromide (M_n_ = 1134 g/mol): ^1^H NMR (400 MHz, CDCl_3_, δ, ppm): 3.35 (br m, 55H, OCH_2_CH_2_CH_2_*CH_2_*Br, O*CH_2_*CH_2_CH_2_*CH_2_*O main chain), 1.88 (t, 4H, OCH_2_CH_2_C*H*_2_CH_2_Br), 1.64 (t, 4H, OCH_2_C*H*_2_C*H*_2_CH_2_O main chain), 1.55 (br m, 47H, OCH_2_C*H*_2_C*H*_2_CH_2_O main chain). ^13^C NMR (75 MHz, CDCl_3_, δ, ppm): 26.5 (OCH_2_C*H*_2_C*H*_2_CH_2_O), 28.3 (OCH_2_C*H*_2_C*H*_2_CH_2_O), 29.7 (OCH_2_C*H*_2_C*H*_2_CH_2_O), 33.6 (OCH_2_CH_2_C*H*_2_CH_2_Br), 69.6 (OCH_2_CH_2_CH_2_C*H*_2_Br), 70.5 (OC*H*_2_CH_2_CH_2_C*H*_2_O).

OTMO-dibromide (M_n_ = 1526 g/mol): ^1^H NMR (400 MHz, CDCl_3_, δ, ppm): 3.35 (br m, 80H, OCH_2_CH_2_CH_2_C*H*_2_Br, OC*H*_2_CH_2_CH_2_C*H*_2_O main chain), 1.88 (t, 4H, OCH_2_CH_2_C*H*_2_CH_2_Br), 1.64 (t, 4H, OCH_2_C*H*_2_C*H*_2_CH_2_O main chain), 1.55 (br m, 72H, OCH_2_C*H*_2_C*H*_2_CH_2_O main chain). ^13^C NMR (75 MHz, CDCl_3_, δ, ppm): 26.4 (OCH_2_C*H*_2_C*H*_2_CH_2_O), 28.3 (OCH_2_C*H*_2_C*H*_2_CH_2_O), 29.7 (OCH_2_C*H*_2_C*H*_2_CH_2_O), 33.6 (OCH_2_CH_2_C*H*_2_CH_2_Br), 69.6 (OCH_2_CH_2_CH_2_C*H*_2_Br), 70.5 (OC*H*_2_CH_2_CH_2_C*H*_2_O).

OTMO-diphtalimide (M_n_ = 1266 g/mol): ^1^H NMR (400 MHz, CDCl_3_, δ, ppm): 7.76 (m, 4H, Pthalimide), 7.63 (m, 4H, Pthalimide), 3.65 (t, 4H, -C*H*_2_CH_2_-Phtalimide), 3.35 (br m, 51H, OC*H*_2_CH_2_CH_2_C*H*_2_O main chain), 1.70 (t, 4H, -CH_2_C*H*_2_-Phtalimide), 1.55 (br m, 51H, OCH_2_C*H*_2_C*H*_2_CH_2_O main chain). ^13^C NMR (75 MHz, CDCl_3_, δ, ppm): 25.3 (OCH_2_C*H*_2_C*H*_2_CH_2_O), 26.4 (OCH_2_C*H*_2_C*H*_2_CH_2_O), 27.0 (OCH_2_C*H*_2_C*H*_2_CH_2_O), 37.7 (OCH_2_CH_2_C*H*_2_CH_2_Phtalimide), 69.9 (OCH_2_CH_2_CH_2_C*H*_2_Phtalimide), 70.4 (OC*H*_2_CH_2_CH_2_C*H*_2_O), 123.0 (Pthalimide), 132.0 (Pthalimide), 133.7 (Pthalimide), 168.2 (Pthalimide).

OTMO-diphtalimide (M_n_ = 1658 g/mol): ^1^H NMR (400 MHz, CDCl_3_, δ, ppm): 7.76 (m, 4H, Pthalimide), 7.63 (m, 4H, Pthalimide), 3.64 (t, 4H, -C*H*_2_CH_2_-Phtalimide), 3.35 (br m, 76H, OC*H*_2_CH_2_CH_2_C*H*_2_O main chain), 1.70 (t, 4H, -CH_2_C*H*_2_-Phtalimide), 1.55 (br m, 76H, OCH_2_C*H*_2_C*H*_2_CH_2_O main chain). ^13^C NMR (75 MHz, CDCl_3_, δ, ppm): 25.3 (OCH_2_C*H*_2_C*H*_2_CH_2_O), 26.4 (OCH_2_C*H*_2_C*H*_2_CH_2_O), 27.0 (OCH_2_C*H*_2_C*H*_2_CH_2_O), 37.7 (OCH_2_CH_2_C*H*_2_CH_2_Phtalimide), 69.9 (OCH_2_CH_2_CH_2_C*H*_2_Phtalimide), 70.5 (OC*H*_2_CH_2_CH_2_C*H*_2_O), 123.0 (Pthalimide), 132.1 (Pthalimide), 133.7 (Pthalimide), 168.2 (Pthalimide).

OTMO-diamines (M_n_ = 1006 g/mol): ^1^H NMR (400 MHz, CDCl_3_, δ, ppm): 3.35 (br m, 51H, OC*H*_2_CH_2_CH_2_C*H*_2_O main chain), 2.87 (br s, 4H, NH_2_), 2.66 (t, 4H, OCH_2_CH_2_C*H*_2_CH_2_NH_2_), 1.55 (br m, 55H, OCH_2_C*H*_2_C*H*_2_CH_2_O main chain). ^13^C NMR (75 MHz, CDCl_3_, δ, ppm): 26.4 (OCH_2_C*H*_2_C*H*_2_CH_2_O, OCH_2_CH_2_C*H*_2_CH_2_NH_2_), 70.4 (OCH_2_CH_2_CH_2_C*H*_2_NH_2_, OC*H*_2_CH_2_CH_2_C*H*_2_O).

OTMO-diamines (M_n_ = 1398 g/mol): ^1^H NMR (400 MHz, CDCl_3_, δ, ppm): 3.35 (br m, 80H, OC*H*_2_CH_2_CH_2_C*H*_2_O main chain, OCH_2_CH_2_C*H*_2_CH_2_NH_2_), 2.67 (br s, 4H, NH_2_), 1.55 (br m, 80H, OCH_2_C*H*_2_C*H*_2_CH_2_O main chain). ^13^C NMR (75 MHz, CDCl_3_, δ, ppm): 26.5 (OCH_2_C*H*_2_C*H*_2_CH_2_O, OCH_2_CH_2_C*H*_2_CH_2_NH_2_), 70.6 (OCH_2_CH_2_CH_2_C*H*_2_NH_2_, OC*H*_2_CH_2_CH_2_C*H*_2_O).

### 3.2. Elemental Analysis

The data obtained in the course of elemental analysis are provided in the Table 3. The closeness of the indices to the theoretical values confirms the structure of the synthesized compounds.

### 3.3. Gel Permeation Chromatography of Functionalized Oligotetramethylene Oxides

In the determination of the molecular mass of compounds, the retention time was from 4.98 to 5.75 min (Figure 5a,b). The small width of the peaks corresponds to the narrow molecular-mass distribution of the oligomer. The obtained values of the average compound molecular mass agree with the theoretical values (Table 4). Furthermore, NMR spectroscopy and GPC revealed that the M_n_ values of compounds in both series remained constant throughout the end-group transformations (Table 4).

### 3.4. FTIR Spectroscopy

#### 3.4.1. FTIR Spectra of Functionalized Oligotetramethylene Oxides

The transformation of the terminal hydroxyls in polyfurites into amino groups was also demonstrated by FTIR spectra (Figure 6).

The absorption band at 665 cm^−1^ (C-Br) is observed in polyfurite spectra after nucleophilic substitution of hydroxy groups with bromine. At the same time, there are no hydroxyl group bands at 3300–3500 cm^−1^, usually present in polyfurite spectra. In the OTMO-diphtalimide spectrum, a characteristic imide peak appears at 1720 cm^−1^. An absorption band of amino groups at 3300–3600 cm^–1^ is observed in the FTIR spectra of OTMO diamines. The rest of the bands of the intermediates and the end product, OTMO-diamine, are identical to those of the initial polyfurite. Thus, it can be concluded that the main chain structure of the oligomer remained unchanged, and only the terminal groups were involved in the reactions.

#### 3.4.2. FTIR Spectra of the Synthesized Elastomers

The FTIR spectra of the synthesized elastomers are shown in Figure 7a,b. The NH band of urethane can be found at 3350 cm^−1^ as a broad absorption. A broad band with the center at 2950 cm^−1^ and the one at 2860 cm^−1^ were assigned to the CH asymmetric stretching and the symmetric one in the CH_2_ groups. The absorption bands at 1542, 1454, and 1412 cm^−1^ were assigned to the amide−NH stretching. The elastomers synthesized from 2,4-toluene diisocyanate (C-1–C-6) have absorption bands at 1600 cm (aryl ring) and also at 1612 cm^−1^. This band is typical for urethane-containing elastomers synthesized on the basis of this diisocyanate. For elastomers synthesized from isophrondiisocyanate, these bands do not appear.

The analysis of the FTIR spectra in the range of carbonyl stretching vibrations (1600–1760 cm^–1^) reveals the important features of the structural organization of the synthesized elastomers. It is known that the microphase separation of soft and hard segments of elastomers with urethane hydroxyl hard blocks is characterized by an absorption band at 1695 cm^−1^ when using isophorone diisocyanate and 1705 cm^−1^ when using 2,4-toluene diisocyanate [43].

The samples synthesized from 2,4-toluylene diisocyanate (Figure 8a,b) in the vibration range of 1600–1760 cm^−1^ have a strong absorption band at 1705 cm^−1^ that corresponds to the hydrogen bond between the hard segments of the elastomer showing the high degree of microphase separation. With an increase in the molecular weight of the oligodiol used in the synthesis of the epoxyurethane oligomer, the intensity of this band decreases. This fact indicates that the degree of microphase separation decreases with an increase in the molecular weight of oligodiol. In the case of using isophorone diisocyanate, the same regularity appears (Figure 9). It should be noted that the degree of microphase separation is higher for samples synthesized from isophorone diisocyanate.

### 3.5. Differential Scanning Calorimetry Data

The thermal properties of the synthesized elastomers were studied by differential scanning calorimetry. First, the samples were heated to 150 °C, then cooled to 100 °C below zero, kept for 30 min, and heated at a heating rate of 5 °C/min. In Figure 10a,b, the reheating thermograms of the samples C1–C12 are shown. The thermophysical properties of the synthesized elastomers are shown in Table 5.

From the presented thermograms, it can be seen that when using the hardener with a molecular weight of 1000, the glass transition temperature is 5–8 °C higher than when using a hardener with a molecular weight of 1400. This is due to an increase in the segmental mobility of polymer chains. On the other hand, the use of a hardener with a higher molecular weight provides the prerequisites for the crystallization of the polymer, which should reduce the segmental activity of the chains.

The elastomers synthesized from isophorone diisocyanate have a glass transition temperature lower by 5–10 °C than elastomers from 2,4-toluene diisocyanate. This is due to the lower degree of microphase separation of soft and hard segments in elastomers (Figure 8a,b and Figure 9a,b). In this case, an increase in the degree of microphase separation leads to a decrease in the degree of crystallinity of elastomers. It should be noted that the melting temperature of the flexible phase of all the elastomers except C-7 (there is no melting in sample C-7) is 29–30 °C.

### 3.6. Deformation and Strength Characteristics

According to the data obtained during mechanical tests (Table 6), the deformation–strength characteristics depend on both the molecular and supramolecular structure of the elastomers.

The elastomers synthesized from 2,4-toluene diisocyanate show a higher Young’s modulus (E_100_), which is explained by a higher degree of crystallinity of the elastomers. However, elastomers synthesized from isophrondiisocyanate, due to a higher degree of microphase separation of soft and hard segments, are characterized by higher strength characteristics. With an increase in the molecular weight of the used hardener, Young’s modulus also increases. Increasing the molecular weight of EUO when using a higher molecular weight OTMO in its synthesis leads to an increase in the deformation characteristics of the cured elastomer.

## 4. Conclusions

For the first time, a method has been developed for the synthesis of oligotetramethylene oxides with terminal amino groups, including the initial bromination of oligotetramethylene oxide diols, followed by the Gabriel reaction.

Six epoxyurethane oligomers were prepared using the oligotetramethylene oxide diol with M_n_ ~ 1008 g∙mol^−1^, M_n_ ~ 1400 g∙mol^−1^, isophorone diisocyanate, 2,4-toluene diisocyanate, and epoxy alcohol-glycidol.

Twelve elastomers from oligomers with urethane hydroxyl hard segments were prepared using synthesized amines.

For the first time, on the basis of epoxyurethane oligomers, synthesized on the basis of polyethers, crystallizable elastomers have been obtained.

The degree of microphase separation is higher for samples synthesized from isophorone diisocyanate.

It has been shown that the use of new oligoamines makes it possible to obtain elastomers with a controlled degree of crystallinity, which allows them to be used as shape memory materials. At the same time, the glass transition temperature of elastomers −60–70 °C allows them to be used in extreme conditions of the far North.

The elastomers synthesized from 2,4-toluene diisocyanate exhibit a higher Young’s modulus (E_100_) due to a higher degree of crystallinity of the elastomers. However, elastomers synthesized from isophrondiisocyanate, due to a higher degree of microphase separation of soft and hard segments, are characterized by higher strength characteristics.

## Figures and Tables

**Figure 1 polymers-14-02136-f001:**
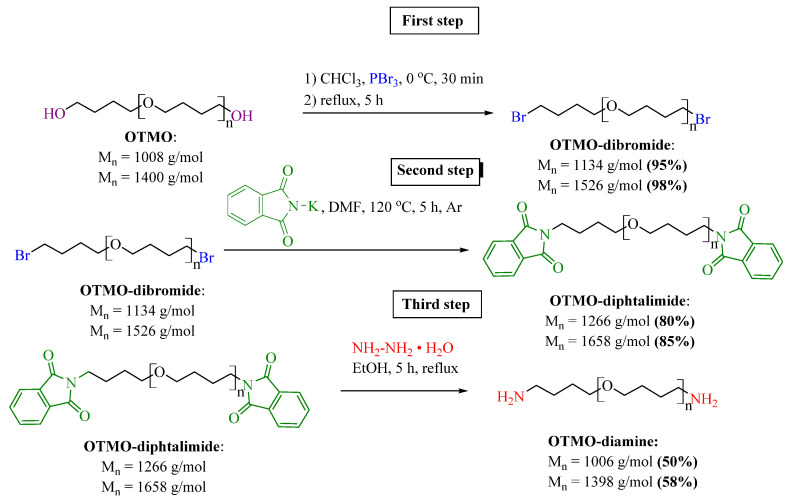
Synthetic route for amino terminated oligo tetramethylene oxide (OTMO-diamine).

**Figure 2 polymers-14-02136-f002:**
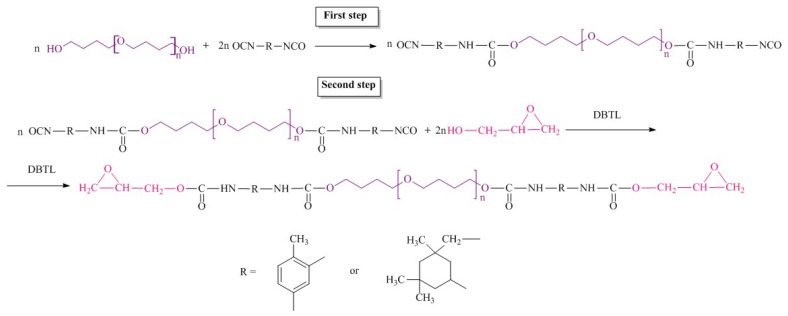
Synthetic route for epoxyurethane oligomers.

**Figure 3 polymers-14-02136-f003:**
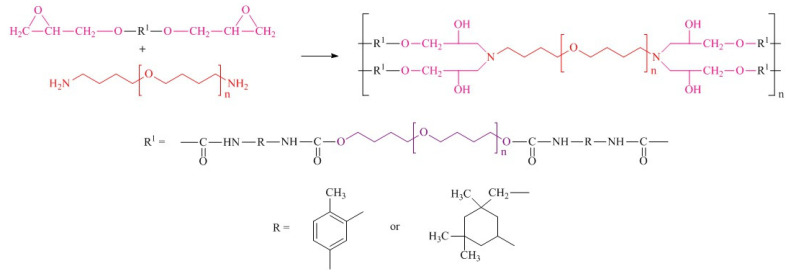
Synthetic route of polymers.

**Figure 4 polymers-14-02136-f004:**
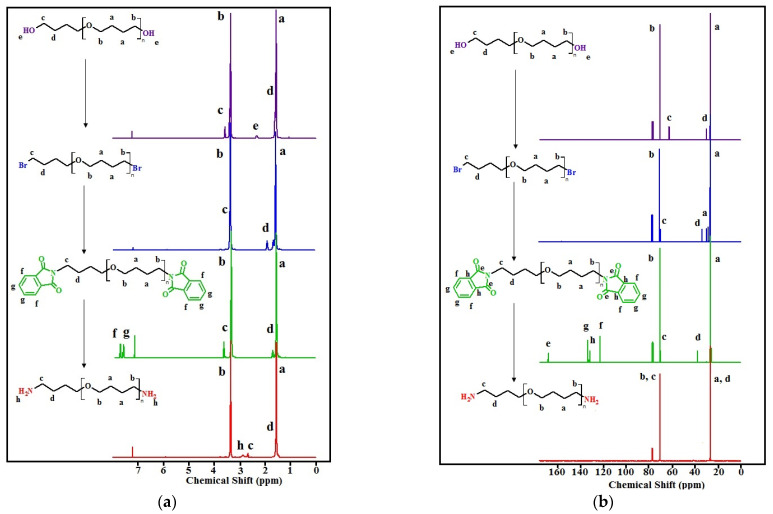
^1^H NMR spectra: (**a**)—with M_n_ ~ 1008 g∙mol^−1^ and its derivatives, (**c**) OTMO with M_n_ ~ 1400 g∙mol^−1^ and its derivatives; ^13^C NMR spectra: (**b**)—with M_n_ ~ 1008 g∙mol^−1^ and its derivatives, (**d**) OTMO with M_n_ ~ 1400 g∙mol^−1^ and its derivatives.

**Figure 5 polymers-14-02136-f005:**
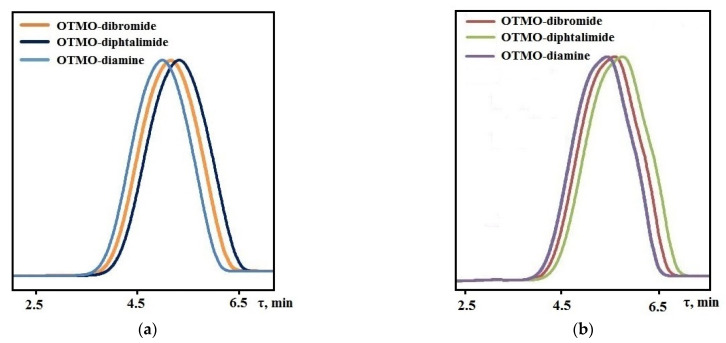
GPC chromatograms: (**a**) OTMO with M_n_ ~ 1008 g∙mol^−1^ and its derivatives; (**b**) OTMO with M_n_ ~ 1400 g∙mol^−1^ and its derivatives recorded using THF as the mobile phase.

**Figure 6 polymers-14-02136-f006:**
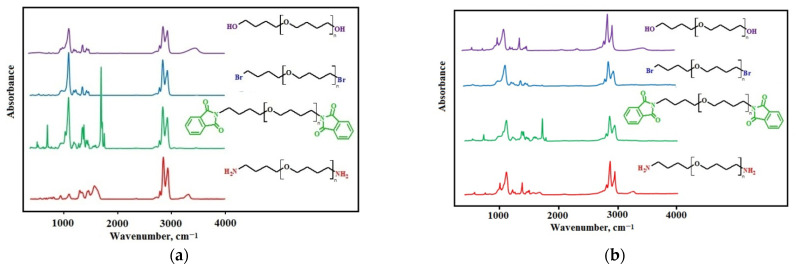
The FTIR spectra for the samples: (**a**) OTMO with M_n_ ~ 1008 g∙mol^−1^ and its derivatives; (**b**) OTMO with M_n_ ~ 1400 g∙mol^−1^ and its derivatives.

**Figure 7 polymers-14-02136-f007:**
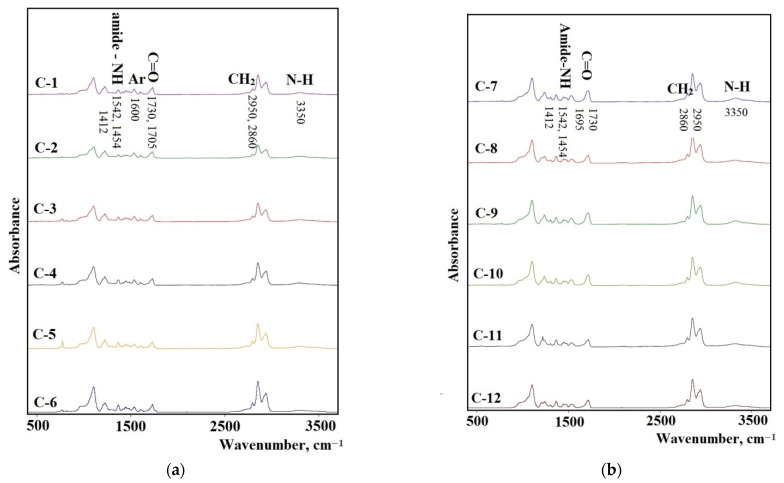
The FTIR spectra for the samples: (**a**) C-1, C-2, C-3, C-4, C-5, C-6; (**b**) C-7, C-8, C-9, C-10, C-11, C-12.

**Figure 8 polymers-14-02136-f008:**
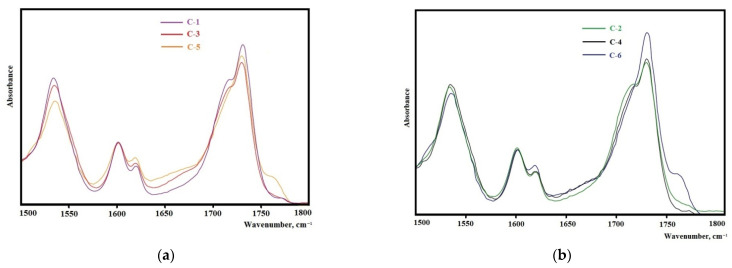
Sections of the FTIR spectra for the samples: (**a**) C-1, C-3, C-5; (**b**) C-2, C-4, C-6.

**Figure 9 polymers-14-02136-f009:**
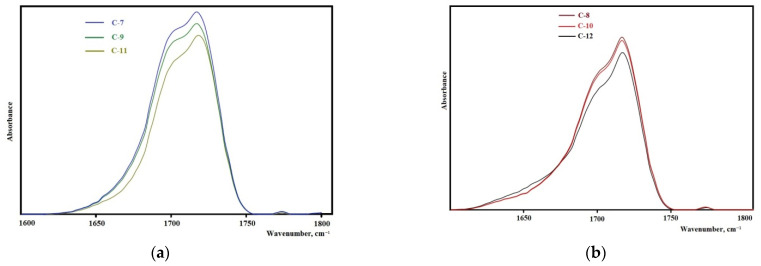
Sections of the FTIR spectra for the samples: (**a**) C-7, C-9, C-11; (**b**) C-8, C-10, C-12.

**Figure 10 polymers-14-02136-f010:**
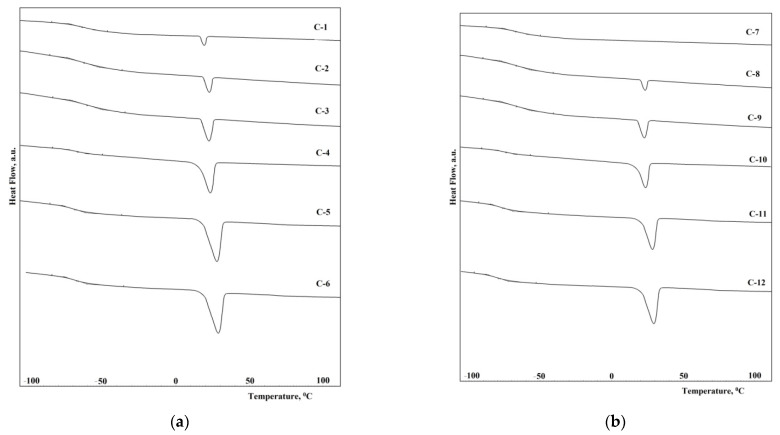
DSC-curves of the epoxy urethane samples from: (**a**) C-1, C-2, C-3, C-4, C-5, C-6; (**b**) C-7, C-8, C-9, C-10, C-11, C-12.

**Table 1 polymers-14-02136-t001:** Composition and properties of the synthesized oligomers.

Product Code	Molecular Weight of Initial EUO OTMO	Diisocyanate Type	Content of Free Isocyanate Groups, wt %		Content of Free Epoxy Groups, wt %	
			Calculated	Determined	Calculated	Determined
FP-1	1008	2,4-toluene diisocyanate	6.23	6.31 ± 0.03	5.82	5.71 ± 0.03
FP-2	1008	isophorone diisocyanate	5.82	5.85 ± 0.03	5.43	5.31 ± 0.03
FP-3	1400	2,4-toluene diisocyanate	4.81	4.92 ± 0.03	4.64	4.72 ± 0.03
FP-4	1400	isophorone diisocyanate	4.56	4.62 ± 0.03	4.37	4.45 ± 0.03
FP-5	2000	2,4-toluene diisocyanate	3.57	3.65 ± 0.03	3.51	3.47 ± 0.03
FP-6	2000	isophorone diisocyanate	3.43	3.53 ± 0.03	3.54	3.45 ± 0.03

**Table 2 polymers-14-02136-t002:** Compositions for preparation of the polymers.

Composition Code	Oligomer Code	Curing Agent
C-1	FP-1	OTMODA 1000
C-2	FP-3	OTMODA 1000
C-3	FP-5	OTMODA 1000
C-4	FP-1	OTMODA 1400
C-5	FP-3	OTMODA 1400
C-6	FP-5	OTMODA 1400
C-7	FP-2	OTMODA 1000
C-8	FP-4	OTMODA 1000
C-9	FP-6	OTMODA 1000
C-10	FP-2	OTMODA 1400
C-11	FP-4	OTMODA 1400
C-12	FP-6	OTMODA 1400

**Table 3 polymers-14-02136-t003:** Elemental analysis of the synthesized compounds.

	C, %		H, %		N, %	
	Founded	Calculated	Founded	Calculated	Founded	Calculated
OTMO-dibromide (M_n_ = 1134 g/mol)	58.41	58.20	9.87	9.70	-	-
OTMO-dibromide (M_n_ = 1526 g/mol)	60.58	60.38	10.19	10.06	-	-
OTMO-diphtalimide (M_n_ = 1266 g/mol)	67.48	67.30	9.49	9.32	2.37	2.21
OTMO-diphtalimide (M_n_ = 1658 g/mol)	67.32	67.15	9.90	9.74	1.84	1.69
OTMO-diamines (M_n_ = 1006 g/mol)	65.75	65.61	11.46	11.33	2.90	2.78
OTMO-diamines (M_n_ = 1398 g/mol)	66.05	65.90	11.40	11.27	2.17	2.00

**Table 4 polymers-14-02136-t004:** Molecular weight characteristics of the compounds determined via ^1^H NMR spectroscopy, and GPC.

	M_n_ ^1^		M_n_ ^2^	
	GPC	^1^H	GPC	^1^H
OTMO-dibromide	1121 (5.105 *)	1134	1480 (5.550 *)	1526
OTMO-diphtalimide	1250 (5.265 *)	1266	1640 (5.748 *)	1658
OTMO-diamines	1020 (4.980 *)	1006	1380 (5.426 *)	1398

M_n_ ^1^—Number-average molecular weight of compounds synthesized on the basis of OTMO with M_n_ ~ 1008 g∙mol^−1^, M_n_ ^2^—Number-average molecular weight of compounds synthesized on the basis of OTMO with M_n_ ~ 1400 g∙mol^−1^, *—Retention time, min.

**Table 5 polymers-14-02136-t005:** Thermophysical properties of synthesized elastomers.

Composition Code	Glass Transition Temperature of the Soft Phase, °C	Melting Temperature of the Soft Phase, °C
C-1	−63	29
C-2	−61	29
C-3	−59	29
C-4	−63	29
C-5	−65	29
C-6	−64	29
C-7	−67	-
C-8	−66	30
C-9	−65	30
C-10	−73	30
C-11	−71	30
C-12	−70	30

**Table 6 polymers-14-02136-t006:** Physical–mechanical characteristics of the synthesized elastomers.

Composition Code	σ_k_, MPa	ε_k_, %	E_100_, MPa
C-1	6.2 ± 0.3	175 ± 7	5.3 ± 0.3
C-2	6.1 ± 0.3	196 ± 7	5.7 ± 0.3
C-3	6.3 ± 0.3	224 ± 7	6.0 ± 0.3
C-4	6.1 ± 0.3	115 ± 5	5.8 ± 0.3
C-5	7.2 ± 0.4	121 ± 5	6.0 ± 0.3
C-6	6.9 ± 0.3	141 ± 5	6.2 ± 0.3
C-7	6.5 ± 0.3	205 ± 7	4.1 ± 0.2
C-8	10.2 ± 0.5	221 ± 7	4.2 ± 0.2
C-9	9.3 ± 0.4	256 ± 10	4.6 ± 0.2
C-10	6.2 ± 0.3	196 ± 7	4.9 ± 0.2
C-11	6.4 ± 0.3	206 ± 7	5.0 ± 0.3
C-12	7.0 ± 0.3	261 ± 10	5.5 ± 0.3

## Data Availability

The most significant data generated or analyzed during this study are included in this published article. Further results obtained during the current study are available from the corresponding author on reasonable request.

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
