# Peer review of "Synthesis and Study of Physical and Mechanical Properties of Urethane-Containing Elastomers Based on Epoxyurethane Oligomers with Controlled Crystallinity"

_polymers, 2022, doi:10.3390/polym14112136_

Round 1
Reviewer 1 Report
This paper described the synthesis of a series of epoxy-polyurethane with an oligodiisocyanate as an intermediate product, and controlled crystallinity and mechanical properties were studied. For the crystallinity was related with the application as shape memory materials, the paper showed significance to develop new type of polyurethane shape memory materials. However, the paper can be accepted after major revision.
- In introduction, the author mainly described the shape memory polymers, while the shape memory performance was not researched in the paper.
- The the shape memory performance of polyurethane was related with not only crystallinity, but also hydrogen bonds. Such as in the published paper of “Polymers, 2019, 11, 1002”
- Introduction, the author described “The difference in the polarity of soft and hard segments results in the microphaseseparation followed by the formation of hard domains.......”, the relevant published papers about the effect of uniform-sized hard segments on the microphase separation should be introduced and cited in the manuscript, such as “Journal of Biomaterials Science, Polymer Edition, 2019, 30: 1212-1226; Materials Science & Engineering C, 109 (2020) 110571”
- In this paper, the crystallinity of elastomers cannot be proved completely. The XRD was suggested to be used to study the crystallinity.
- In my opinion, the influence of EUO OTMO molecular weight on mechanical properties is crucial, but the explanation in this paper is not sufficient.
Author Response
Dear Reviewer! Thank you greatly for the careful work with our paper. We have tried to consider all the remarks You have made in the report.

Reviewer 2 Report
- In page 13, no need to have the following the sentence in the manuscript. "This section is not mandatory but can be added to the manuscript if the discussion is 339 unusually long or complex."
- The authors only provide "molecular weight" in the manuscript. Which molecular weight did the author refer to ? Mn or Mw?
- In the manuscript, the authors only provide Mn. How about the Mw and PDI?
Author Response
Dear Reviewer! Thank you greatly for the careful work with our paper. We have tried to consider all the remarks. You have made in the report.

Reviewer 3 Report
The influence of the molecular weight of oligoamine, oligoether, and the type of diisocyanate on the physical and mechanical properties of elastomers with urethane hydroxyl hard segments was studied. The paper is interesting and could be published after the revision.-Literature review is very broad. It should be more concentrated and just describe what new will be developed in field of the materials and what problems of the materials will be solved.- Synthesis of OTMO-diamine is long and complicated if we would think about industrial product. The authors should demonstrate why this diamine will be better in production of polyurethanes as compared with industrial and cheap diamines ?- How were Mn of products in Figure 1 measured ? Was this done by GPC ?-It seems that in Figure 2 in first step also polymers\oligomers could be formed, not only the derivative with 2 diisocyanate groups. Could molecular weight of the product measure by GPC or mass spectrometry ?- This “This section is not mandatory but can be added to the manuscript if the discussion is unusually long or complex“ should be removed from the conclusions.-Mw should be also presented for oligomers and polymers.-Advantages and disadvantages of the new polymers should be exactly described in conclusions and compared with that of other known elastomeric polyurethanes.
Author Response

(The authors gave the same response as above.)

Round 2
Reviewer 1 Report
The authors has followed all the remarks given by reviewers and made significant quality improvement. Therefore, I recommend to be published in current form.Reviewer 3 Report
If editor and other reviewers agree I could also recommend the paper for publication after the revision.